# Acupuncture for constipation in patients with stroke: protocol of a systematic review and meta-analysis

Jingbo Zhai,[1] Wei Mu,[2] Jinhua Si,[3] Yan Li,[4] Chen Zhao,[5] Hongcai Shang,[6] Huanan Li,[7,8] Guihua Tian[6]

## ABSTRACT

**Introduction** Constipation is one of the most common complications in patients with stroke. Acupuncture has gained increased popularity for the management of constipation. However, there is a lack of supportive evidence on the efficacy of acupuncture for poststroke constipation. This systematic review aims to collect and critically appraise all the available evidence about the efficacy and safety of the acupuncture for constipation in poststroke patients.

**Methods and analysis** A comprehensive search of Pubmed, Embase, Cochrane Central Register of Controlled Trials, Web of Science, four Chinese databases (National Knowledge Infrastructure (CNKI), Chinese Biomedical Literatures database (CBM), Wanfang Digital Periodicals (WANFANG) and Chinese Science and Technology Periodicals (VIP) database), one Japanese medical database (National Institute of Informatics, CiNii) and one Korean medical database (Oriental Medicine Advanced Searching Integrated System, OASIS) will be conducted to identify randomised controlled trials of acupuncture for constipation in poststroke patients. There is no restriction on language or publication status. The primary outcome measure will be frequency of bowel movement. The risk of bias will be assessed using the approach recommended by Cochrane Handbook for Systematic Reviews of Interventions. We will conduct the meta-analysis to synthesise the evidence for each outcome, if possible. The heterogeneity will be statistically assessed using a $\chi^2$ test and $I^2$ statistic. This protocol is developed following the guideline of Preferred Reporting Items for Systematic Reviews and Meta-analyses Protocols 2015.

**Ethics and dissemination** The ethical approval is not required because no primary data are collected. The findings will be presented at scientific conferences or a peer-reviewed scientific journal.

**PROSPERO registration number** CRD42017076880.

For numbered affiliations see end of article.

**Correspondence to**
Dr Huanan Li;
lihuanan1984@126.com and
Prof. Guihua Tian;
Rosetgh@163.com

## Strengths and limitations of this study

► This review will provide a comprehensive assessment regarding the effect of acupuncture for constipation in patients with stroke.
► Only randomised controlled trials will be included which are more likely to provide unbiased information than other study designs.
► The reliability of the results will largely depend on the comprehensiveness and the methodological quality of the primary studies included in this review.

Constipation is one of the most common complications in patients with stroke.[3] The evidence from a previous systematic review shows that the incidence of constipation in patients with stroke is 48% (95% CI 33% to 63%).[4] The incidence of constipation in patients with haemorrhagic stroke is higher than that in patients with ischaemic type.[4] The incidence in rehabilitation stage is higher than that in acute stage.[4]

Constipation has a negative impact on the patient's psychological well-being and physical health, restricts social activities, reduces quality of life, contributes to a poor outcome and is associated with high healthcare costs.[4–6]

Stool softeners, prokinetic agents, osmotic and stimulant laxatives and lifestyle or dietary modification are common treatments for constipation.[7] Conventional treatments may be associated with unwanted side effects, such as bloating, dehydration, a high recurrence rate after ceasing drugs.[7 8] Most of patients with chronic constipation are not satisfied with current treatment options in Europe.[9] Many patients with constipation seek help from alternative therapies.[9]

Acupuncture, as an important part of complementary and complementary medicine, has gained increased popularity for the management of constipation in Western countries.[9] There are some possible mechanisms of acupuncture for constipation.[8]

## INTRODUCTION

Stroke is one of the leading causes of death and disability worldwide.[1] The incidence of stroke is subject to large variation globally.[2] The overall incidence in low to middle-income countries exceeded that in high-income countries by 20% from 2000 to 2008.[2]

Acupuncture can promote intestinal canal peristalsis through contacting the intestinal wall and regulating nervous and body fluids.[8] Acupuncture can increase rectal internal pressure to restore the defecation sense by stimulating parasympathetic nerve.[8]

A systematic review shows that acupuncture is safe for chronic functional constipation.[10] A randomised trial supports the use of electroacupuncture for chronic severe functional constipation.[11] However, whether the evidence is transferable to the stroke population remains unclear.

The recent studies mainly focus on the incidence of constipation after stroke.[4 5] The management strategies for constipation in patients with stroke remain poorly investigated.[3]

A 2014 systematic review evaluated the efficacy and safety of acupuncture and moxibustion for poststroke constipation.[12] This review had evident flaws that threatened the authenticity of their findings. First, meta-analysis found that acupuncture and moxibustion were significantly more effective than other treatments (OR=2.10, 95% credible interval 1.25 to 3.54, p=0.005) for constipation in patients with stroke. Acupuncture and moxibustion were addressed as a whole, and the efficacy of acupuncture alone was not systematically investigated, despite that five of the eight included trials compared acupuncture alone with another treatment. Second, the methodological quality of eight included articles was very poor. Third, the total effective rate, a subjective outcome measure, was chosen as the primary outcome. Fourth, the sample size was small. To our knowledge, several new randomised controlled trials (RCTs) have been published since the meta-analysis was published.[13 14] Overall, there is a lack of supportive evidence on the efficacy and safety of acupuncture for constipation in patients with stroke.

The aim of this study is to systematically review current available literature to assess the efficacy and safety of the acupuncture treatment for post-stroke constipation.

## METHODS

This protocol is developed following the guideline of Preferred Reporting Items for Systematic Reviews and Meta-analyses Protocols (PRISMA-P) 2015.[15]

### Inclusion criteria
#### Types of studies
We will only include RCTs which are more likely to provide unbiased information than other study designs.[16] We will exclude quasirandomised RCTs, such as those allocating by alphabetical order, alternate days of the week or date of birth. Cross-over trials will be excluded because of potential for a carry-over effect. There is no restriction on language or publication status.

#### Types of participants
We will include adults (over 18 years old) suffering from constipation after a first or recurrent stroke. We also consider RCTs in which a prior history of constipation before the stroke diagnosis is not investigated but excluded trials reporting on patients with a history of constipation before the stroke diagnosis. Stroke is defined as 'rapidly developed signs of focal or global disturbance of cerebral function, lasting more than 24 hours or leading to death with no apparent cause other than that of vascular origin', according to WHO criteria.[17] We will include patients with stroke irrespective of any type (ischaemic or haemorrhagic) or phase (acute, subacute or chronic). Acute and subacute stroke is defined as less than 6 months since onset, and chronic stroke lasts more than 6 months since onset.[18]

All of patients should be diagnosed as constipation according to at least one of the current or past definitions or guidelines of constipation, such as Rome II/III diagnostic criteria or guidelines for clinical research on Chinese new herbal medicine.[12]

There is no restriction on age, sex or ethnicity of the enrolled subjects.

### Types of interventions
#### Experimental interventions
We will include trials using either traditional or contemporary acupuncture. Traditional acupuncture refers to needles inserted in classical meridian points.[19] Contemporary acupuncture refers to needles inserted in non-meridian or trigger points regardless of the source of stimulation (for example, hand, electrical stimulation or fine needle).[19] We will exclude trials in which treatment without needling, such as acupressure, tap-pricking, point injection and laser acupuncture. No restrictions are imposed on times of treatment and length of treatment period.

#### Comparator interventions
The control interventions could be placebo acupuncture, sham acupuncture, no treatment, another active treatment or medication.

Placebo acupuncture refers to a needle attached to the skin surface without penetrating the skin.[20]

Sham acupuncture is defined as a needle placed in an area close to but not in acupuncture points or subliminal skin electrostimulation via electrodes attached to the skin.[20]

We consider another active treatment or medication to be pharmacological and non-pharmacological treatment or medication, such as laxatives, emollients, lubricants, lifestyle or dietary modification.

We will investigate the comparisons listed below:
1. Acupuncture only compared with no treatment;
2. Acupuncture only compared with placebo or sham treatment;
3. Acupuncture plus another active treatment or medication compared with another active treatment or medication alone;
4. Acupuncture plus another active treatment or medication compared with placebo or sham treatment plus another active treatment or medication.

## Types of outcome measures
### Primary outcome
The primary outcome measure will be frequency of bowel movement.[21 22] Bowel movement frequency is defined as the mean number of spontaneous bowel movements per week.[23]

### Secondary outcome
Secondary outcome measures include proportion of patients with stool consistency, proportion of patients using rescue medication such as laxatives or rectal evacuants, quality of life (QoL) and mean transit time.

Stool consistency will be defined by trialists or measured based on scales such as Bristol Stool Form Scale.[24]

QoL will be measured by generic or condition-specific scales, such as Short Form 36 Health Surveys.[25]

Transit time is defined as the time from the first perception of wanting to defecate to the end of defecation.[26]

We will sum up the number of adverse events (AEs) and calculate the proportion of AEs.[23]

## Search strategy
### Electronic searches
The published literature will be identified by searching Pubmed, Embase, Cochrane Central Register of Controlled Trials and Web of Science. Four Chinese databases (CNKI, CBM, WANFANG and VIP database), one Japanese medical database (CiNii) and one Korean medical database (OASIS) will also be systematically searched to identify any relevant study.

The search strategy is developed by a medical librarian (JS) according to key terms from previous literature reviews.[27 28] The detailed search strategy is attached (see online supplementary appendix 1). The terms will be modified as necessary for other databases. We will not apply any language or date restrictions.

### Searching other resources
We will check the reference lists of identified relevant RCTs and reviews for additional studies. We will contact experts in the field of stroke and constipation to identify any additional trials.

The WHO International Clinical Trials Registry Platform (ICTRP), ClinicalTrials.gov will also be checked to identify planned, ongoing or unpublished trials. Google Scholar will be searched to identify any grey literature.

## Data collection and analysis
### Selection of studies
Two review authors (WM and JS) will independently assess abstracts and titles of studies identified by literature search. Duplicates will be omitted using EndNote software (V.X7.0).[29] Relevant studies will be selected against the predefined inclusion criteria. If necessary, reviewers will examine full-text reports to identify eligible studies. EndNote software will also be used to manage records. We will illustrate the selection process in a PRISMA diagram.[30] Any disagreement will be resolved by consensus.

## Data extraction and management
Two review authors (YL and CZ) will independently extract data from the included studies. Calibration exercises will be conducted to ensure consistency across reviewers before starting the review. The following information will be extracted using a predetermined data form: general information (title, authors, country of study, funding, year of publication, registry number); details of study (aim, design, inclusion and exclusion criteria, method of randomisation and allocation); study population (age, sex, sample size, number for analysis, type of stroke, phase of stroke); intervention characteristics (type, duration, dose, follow-up time points, compliance); outcome (primary and secondary outcomes, time points, method of outcome assessments, blinding of outcome assessment, adverse effects).

It is possible that patients in different phases of stroke were enrolled in one trial. Whenever necessary the authors of the original trial will be contacted for additional information and clarification of the data. Any disagreement will be resolved by consensus or consultation with a third review author.

## Assessment of risk of bias in included studies
Two authors (HL and HS) will independently assess the risk of bias using the approach recommended by Cochrane Handbook for Systematic Reviews of Interventions.[31] The following risk of bias domains will be assessed: sequence generation (selection bias); allocation concealment (selection bias); blinding of participants and personnel (performance bias); blinding of outcome assessment (detection bias); incomplete outcome data (attrition bias); selective outcome reporting (reporting bias) and other bias.

We will attempt to describe what is reported to have happened in each study for each domain of risk of bias. Thus, we will be able to provide the rationale for the judgement of whether this domain is at low, high or unclear risk of bias. Where necessary, we will contact authors of included studies for missing information or clarification.

If all domains are at low risk of bias, the overall risk of bias of individual studies will be categorised as low risk of bias. Otherwise, overall risk of bias will be categorised as high risk of bias.[32] The 'risk of bias' summary will be presented graphically.

## Measures of treatment effect
We will use the risk ratio with 95% CI to express the estimate of the effect for dichotomous outcome.

For continuous outcome, we will express the estimate of the effect as mean difference with 95% CI. When the same outcome is measured in a variety of ways, the standardised mean difference with 95% CI will be used to express the size of the intervention effect.

## Dealing with missing data
We will attempt to contact study authors for missing data or clarification, where feasible. The following strategies

will be used to evaluate the potential influence of missing data.[32]

1. Worst-case scenario analysis: all participants with missing data counted as failures.
2. Extreme worst-case analysis: participants with missing data in experimental group counted as failures and participants with missing data in control group counted as successes.
3. Extreme best-case analysis: participants with missing data in experimental group counted as successes and participants with missing data in control group counted as failures.

### Assessment of heterogeneity

We will assess the statistical heterogeneity using a $\chi^2$ test.[33] In addition, we will quantify heterogeneity using the $I^2$ statistic value which ranges from 0% to 100%.[34] $P<0.1$ of $\chi^2$ test or $I^2>50\%$ indicates statistically significant heterogeneity.[33 34] Potential clinical heterogeneity will be assessed by prespecified subgroup analyses.

### Assessment of reporting biases

When a meta-analysis includes 10 or more RCTs, we will assess asymmetry using funnel plots visually.[32] In addition, we will test asymmetry using the Harbord modified test for dichotomous outcomes and Egger test for continuous outcomes.[32]

### Data synthesis

We will combine more than one trial to estimate pooled intervention effect using the meta-analysis when studies examine the same intervention and outcomes with comparable methods in similar populations.[31]

We will pool the continuous data using the inverse variance method and dichotomous data using the Mantel-Haenszel method.[31]

We will use the fixed-effect model to combine data when statistical heterogeneity is low. However, when $P<0.1$ or $I^2>50\%$, the random-effect model will be used to provide a more conservative estimate of effect.[35]

All analyses will be conducted with Review Manager V.5.3 software. If a meta-analysis is not possible, we will provide a narrative summary of the results from individual studies.

### Subgroup analysis and investigation of heterogeneity

We will perform the following subgroup analyses to investigate heterogeneity when sufficient data are available. We will conduct subgroup analyses based on age, sex, type of stroke (haemorrhagic and ischaemic stroke), different definitions of constipation, phase of stroke (acute, subacute or chronic), type of acupuncture (manual acupuncture, electroacupuncture, etc) and type of control group (placebo, sham acupuncture, no treatment or another active treatment or medication). A subgroup analysis based on population with different diet habits is under plan considering that diet habits may play an important role in constipation development.

The intervention effect will be analysed using the $\chi^2$ test, with $p<0.05$, indicating statistically significant differences between subgroups.

### Sensitivity analysis

We will perform sensitivity analyses to evaluate the robustness of the pooled results excluding trials with high risk of bias, trials in which a prior history of constipation before the stroke diagnosis is not investigated and the option of using missing data (worst-case scenario analysis, extreme worst-case analysis or extreme best-case analysis).[32 34]

### Summary of findings table

We will prepare 'summary of findings' tables including a grade of the overall quality of the body of evidence for each outcome using Grading of Recommendations Assessment, Development and Evaluation profiler (GRADEpro) Guideline Development Tool.[36]

Two review authors will independently assess the quality of the body of evidence according to five GRADE criteria: study limitations, imprecision, inconsistency, indirectness and publication bias. It will fall into one of four possible ratings (high, moderate, low and very low). Any discrepancy will be resolved by consensus or consultation with a third review author.

### Amendments

We will provide the date of any amendment, a description of the change and the rationale in the event of protocol amendments.

## ETHICS AND DISSEMINATION

Ethical approval is not required because no primary data are collected.

This review will provide a comprehensive assessment regarding the effect of acupuncture for constipation in patients with stroke. The results will be fundamental for reliable recommendations in the management of post-stroke constipation.

We will present findings from this systematic review at scientific conferences and publish the findings in a peer-reviewed scientific journal according to the PRISMA guidelines.

**Author affiliations**
[1]Research institute of Traditional Chinese Medicine, Tianjin University of Traditional Chinese Medicine, Tianjin, China
[2]Department of Clinical Pharmacology, The Second Affiliated Hospital of Tianjin University of Traditional Chinese Medicine, Tianjin, China
[3]Library, Tianjin University of Traditional Chinese Medicine, Tianjin, China
[4]School of Nursing, Tianjin University of Traditional Chinese Medicine, Tianjin, China
[5]Hong Kong Chinese Medicine Clinical Study Centre, Hong Kong Baptist University, Hong Kong, China
[6]Key Laboratory of Chinese Internal Medicine of Ministry of Education and Beijing, Dongzhimen Hospital, Beijing University of Chinese Medicine, Beijing, China
[7]First Teaching Hospital of Tianjin University of Traditional Chinese Medicine, Tianjin, China
[8]Laboratory for Biological Effects of Tuina, State Administration of Traditional Chinese Medicine, Tianjin, China

**Contributors** JZ, HL and GT conceived the study; these three authors provided general guidance to the drafting of the protocol. JZ and WM drafted the protocol. JS designed the search strategy. JZ, YL, CZ and HS drafted the manuscript. JZ, WM, JS, YL, CZ, HL, GT and HS reviewed and revised the manuscript. All authors have read and approved the final version of the manuscript.

**Funding** This study is supported by the National Natural Science Foundation of China (grant numbers 81373762, 81603495, 81703936) and Beijing Nova Program (grant number xx2014B049).

**Competing interests** None declared.

**Patient consent** Not required.

**Provenance and peer review** Not commissioned; externally peer reviewed.

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
