## [Reviewer comments · BMJ Open]

ARTICLE DETAILS

TITLE (PROVISIONAL)	Acupuncture for constipation in stroke patients: protocol of a systematic review and meta-analysis
AUTHORS	Zhai, Jingbo; Mu, Wei; Si, Jinhua; Li, Yan; Zhao, Chen; Shang, Hongcai; Li, Huanan; Tian, Guihua

VERSION 1 – REVIEW

REVIEWER	Seungwon Kwon Department of Cardiology and Neurology, College of Korean Medicine, Kyung Hee University
REVIEW RETURNED	16-Dec-2017

GENERAL COMMENTS	Thanks for giving me the opportunity to review interesting protocol manuscript. This paper adequately describes. However, I think authors need to consider some modifications before publication. General 1. Although authors did not want to limit the language in the literature search, Japanese and Korean database are missing. I think it is also necessary to search databases such as CiNii from Japan and OASIS from Korea Methods Types of participants “Trials will be excluded in which patients have a prior history of constipation before the stroke diagnosis” -> If there is an RCT that has not been examined for existing constipation, does it not include it? Investigation may not be possible. This need to be considered. Subgroup analysis and investigation of heterogeneity -> In this study, authors will investigate various acupuncture types including manual acupuncture, electroacupuncture, and hand acupuncture etc.. However, there is no analysis plan for each intervention type in the subgroup analysis. It is also necessary to consider the analysis. Appendix In the Appendix, only pubmed search strategy is included. The search strategy for the rest of the Chinese database should also be provided.
---

REVIEWER	Stefano Ricci Director, UO Neurologia, Local Health Unit 1 of Umbria Region, Italy
REVIEW RETURNED	04-Jan-2018

GENERAL COMMENTS	This is an interesting protocol describing a SR on a common problem we stroke doctors are faced with, that is constipation in stroke patients. I have just a few minor suggestions to Authors: 1) On introduction, Authors mention a review dated 2014, and correctly describe its limits. I suggest to add some information on the results (especially the confidence intervals!) of this review. 2) In the introduction, Authors mention 2 new trials on the subject, both published in Chinese Journals. Is there any paper or presentation from western world Authors? If not, do Authors think that the topic should be better studied in Europe and other western Countries? In fact, different diet habits may play an important role in the development of constipation 3) When describing types of participants, could Authors better separate the acute phase of stroke (i.e. < 7 days) from the other periods? This is relevant because in several Countries patients are admitted in a Stroke Unit for a few days (7-10) and then sent to different facilities, and the way the problem arises in the Stroke Unit is very relevant.
--

VERSION 1 – AUTHOR RESPONSE

The changes we made were colored in red.

Editorial Requirements:

- Please revise the Strengths and Limitations section (after the abstract) to focus on both the methodological strengths and limitations of your study.

Response:

Thank you for your suggestion. We have made changes accordingly to the “Strengths and limitations” section on page 2.

- Please include an ethics and dissemination section in the main text of the manuscript.

Response:

Thank you for your suggestion. We have made changes as required in the “Ethics and dissemination” section on page 8.

Reviewer(s)' Comments to Author:

Reviewer: 1

General

1. Although authors did not want to limit the language in the literature search, Japanese and Korean database are missing. I think it is also necessary to search databases such as CiNii from Japan and OASIS from Korea

Response:

Thank you for your suggestion. We have included the two recommended databases in our literature search plan and made changes in the “ABSTRACT” and “Electronic search” section on page 2 and 5.

Methods

Types of participants

“Trials will be excluded in which patients have a prior history of constipation before the stroke diagnosis”

-> If there is an RCT that has not been examined for existing constipation, does it not include it? Investigation may not be possible. This need to be considered.

Response:

Thank you for your advice. We also consider RCTs in which a prior history of constipation before the stroke diagnosis is not investigated.

If possible a sensitivity analysis will be conducted to evaluate the robustness of the pooled results including and excluding trials in which a prior history of constipation before the stroke diagnosis is not investigated.

We have made revisions to the "Types of participants" section on page 4 and the "Sensitivity analysis" section on page 8.

Subgroup analysis and investigation of heterogeneity

-> In this study, authors will investigate various acupuncture types including manual acupuncture, electroacupuncture, and hand acupuncture etc.. However, there is no analysis plan for each intervention type in the subgroup analysis. It is also necessary to consider the analysis.

Response:

Thank you for your advice. We will perform subgroup analyses based on various types of acupuncture (manual acupuncture, electroacupuncture, etc.). We have made changes in the "Subgroup analysis and investigation of heterogeneity" section on page 8.

Appendix

In the Appendix, only pubmed search strategy is included. The search strategy for the rest of the Chinese database should also be provided.

Response:

Thank you for your advice. We have provided the search strategy for all the Chinese databases including CNKI, CBM, WANFANG, and VIP database in appendix 1.

Reviewer: 2

1) On introduction, Authors mention a review dated 2014, and correctly describe its limits. I suggest to add some information on the results (especially the confidence intervals!) of this review.

Response:

Thank you for your advice. We have provided more information on the results of the previous review in the "introduction" section on page 3.

2) In the introduction, Authors mention 2 new trials on the subject, both published in Chinese Journals. Is there any paper or presentation from western world Authors? If not, do Authors think that the topic should be better studied in Europe and other western Countries? In fact, different diet habits may play an important role in the development of constipation

Response:

Thanks for this useful comment. We are by far unaware of any published report on this topic from western world authors, as the planned systematic search has not been kicked off.

Post-stroke constipation is a global concern. A rigorously-conducted high-impact RCT found that acupuncture is effective for functional constipation (Liu Z, Yan S, Wu J, et al. Acupuncture for Chronic Severe Functional Constipation: A Randomized Trial. *Ann Intern Med* 2016, 165(11):761-769.). We expect the evidence is transferrable to stroke patients with constipation. The topic needs to be studied in both western and eastern countries.

Truely, different diet habits may play an important role in the development of constipation. And it may differ between western and eastern countries. In view of this, we plan to perform subgroup analyses based on population with different diet habits, and added this plan to the "Subgroup analysis and investigation of heterogeneity" section on page 8.

We expect this systematic review to shed light on new perspectives on post-stroke constipation management, thus helping doctors and patients, from both the west and the east, with their clinical decision-making.

3) When describing types of participants, could Authors better separate the acute phase of stroke (i.e. < 7 days) from the other periods? This is relevant because in several Countries patients are admitted in a Stroke Unit for a few days (7-10) and then sent to different facilities, and the way the problem arises in the Stroke Unit is very relevant.

Response:

Thanks for your useful information. We will extract data on the phase of stroke investigated from eligible primary studies, and consider it in our further analysis. It is possible patients in different phases of stroke were enrolled in one trial. In this case, the authors of the original will be contacted for additional data on efficacy and safety categorised by disease periods, if necessary. We have made changes to the “Data extraction and management” section on page 6.

The pathogenesis of constipation may vary with different phases of stroke. Again, we will conduct subgroup analyses based on the phase of stroke (acute, subacute or chronic) when sufficient data are available. Revisions have been made to the “Subgroup analysis and investigation of heterogeneity” section on page 8. We will discuss these topics in the final systematic review report.

VERSION 2 – REVIEW

REVIEWER	Stefano Ricci UO Neurologia USL Umbria 1 Città di Castello (PG) Italy
REVIEW RETURNED	24-Jan-2018
GENERAL COMMENTS	I wish to thank Authors for having kindly answered my questions. I have no further point to discuss.
REVIEWER	Seungwon Kwon Department of Cardiology and Neurology, College of Korean Medicine, Kyung Hee University
REVIEW RETURNED	04-Mar-2018
GENERAL COMMENTS	Thank you for giving me a chance to review this article. I think authors revise this manuscript according to the reviewers' suggestion. Therefore, I think this manuscript could be accepted.